# Biosynthesis of Putrescine from L-arginine Using Engineered *Escherichia coli* Whole Cells

**Hongjie Hui, Yajun Bai, Tai-Ping Fan, Xiaohui Zheng and Yujie Cai *** 

The Key Laboratory of Industrial Biotechnology, Ministry of Education, School of Biotechnology, Jiangnan University, 1800 Lihu Road, Wuxi 214222, China; rockyhui@foxmail.com (H.H.); baiyj@nwu.edu.cn (Y.B.); tpf1000@cam.ac.uk (T.-P.F.); zhengxh318@nwu.edu.cn (X.Z.)

**\*** Correspondence: yjcai@jiangnan.edu.cn; Tel.: +86-18961727911

**Abstract:** Putrescine, a biogenic amine, is a highly valued compound in medicine, industry, and agriculture. In this study, we report a whole-cell biocatalytic method in *Escherichia coli* for the production of putrescine, using L-arginine as the substrate. L-arginine decarboxylase and agmatine ureohydrolase were co-expressed to produce putrescine from L-arginine. Ten plasmids with different copy numbers and ordering of genes were constructed to balance the expression of the two enzymes, and the best strain was pACYCDuet-*speB-speA*. The optimal concentration of L-arginine was determined to be 20 mM for this strain. The optimum pH of the biotransformation was 9.5, and the optimum temperature was 45 °C; under these conditions, the yield of putrescine was 98%. This whole-cell biocatalytic method appeared to have great potential for the production of putrescine.

**Keywords:** L-arginine; putrescine; *Escherichia coli*; whole-cell catalysis; co-expression

## 1. Introduction

Putrescine, also known as 1,4-butanediamine, is an aliphatic diamine with a very unpleasant smell and is one of the biogenic amines [1]. In 1889, it was first isolated from *Vibrio cholera*, and its common name was derived from its occurrence in rotten meat [2]. In both prokaryotic and eukaryotic cells, putrescine is an essential regulator, which affects cell growth, differentiation, proliferation, and various physiological processes [3,4]. In mammals, putrescine regulates the intestinal flora and improves the intestinal immune function [4]. Putrescine is also an important metabolite of the human intestinal flora, which is crucial to their life processes [5]. Treating plants with putrescine positively affects their growth, productivity, and stress tolerance [6]. In agriculture, putrescine can enhance salt-tolerance and extend the shelf life of vegetable crops [7,8]. In the plastic industry, putrescine is a co-monomer with adipic acid, to prepare the high-quality industrial plastic nylon 46, which has excellent solvent resistance and mechanical properties [9–11].

The biological synthesis methods of putrescine have been of great research interest in the past decades around the world. *Escherichia coli*, modified by gene knockout and stronger promoter replacement, achieved a space-time yield of 0.75 g $L^{-1}$ $h^{-1}$ with glucose as the substrate, in a high-density culture [10]. Ornithine decarboxylase was successfully modified to achieve a higher catalytic activity, by rational design and molecular docking [12]. *Corynebacterium glutamicum* was engineered to construct a high-yielding strain (NA6) through gene knockout and metabolic regulation, with L-arginine as the substrate [13–16]. Putrescine has been synthesized in *C. glutamicum,* using xylose as the substrate [17]. Putrescine was synthesized from agmatine, via the agmatine deiminase pathway by *Lactococcus lactis* [18]. A combination of various putrescine synthesis pathways in *Saccharomyces cerevisiae*, with glutamic acid as the substrate, achieved a putrescine concentration of 86 mg $L^{-1}$ after 48 h [19].

Multienzyme cascade reactions have been widely used to convert cheap substrates into highly valued products. This is a more economical and effective approach than de novo synthesis with glucose under some circumstances [20–22]. In this study, one metabolic pathway for the production of putrescine was constructed using L-arginine as the substrate. In *E. coli*, L-arginine decarboxylase (ADC) and agmatine ureohydrolase (AUH) were co-expressed. In the first reaction, L-arginine was decarboxylated by ADC to form agmatine and $CO_2$. In the second reaction, AUH removed urea from agmatine to produce putrescine (Figure 1). *E. coli* was chosen as the whole-cell catalyst for its excellent protein expression and well-established gene manipulation.

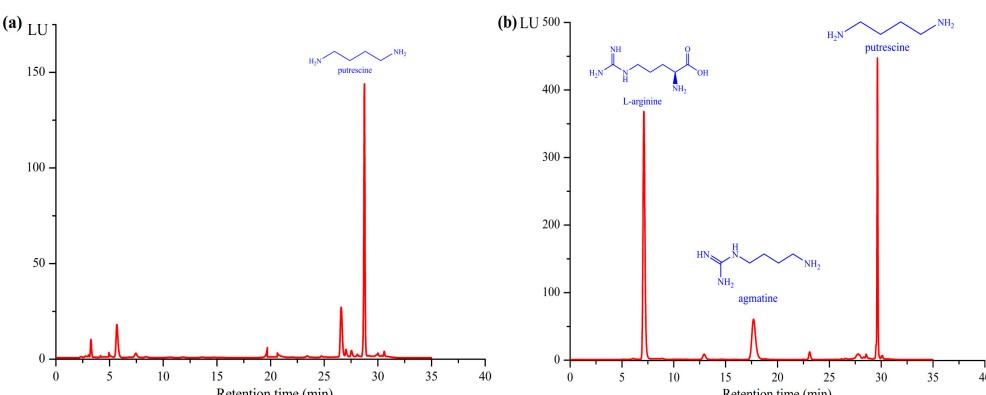

**Figure 1.** Scheme for the production of putrescine. L-arginine is decarboxylated by ADC to form agmatine and $CO_2$. AUH then removes urea from agmatine to produce putrescine. ADC: L-arginine decarboxylase, encoded by *speA*; AUH: agmatine ureohydrolyase, encoded by *speB*.

## 2. Results

### 2.1. Construction of E. coli Whole-Cell Biocatalysts

First, 10 plasmids were constructed using five plasmids with different copy numbers and different orders of *speA* and *speB*. The 10 constructed strains were used as whole-cell biocatalysts to convert L-arginine to putrescine, and each reaction mixture was derivatized with o-phthaldialdehyde and monitored by high performance liquid chromatography (HPLC). A new peak having the same retention time (~30 min) as standard putrescine appeared in the reaction mixture, which was confirmed to be the product putrescine (Figure 2). Sodium dodecyl sulfate polyacrylamide gel electrophoresis (SDS-PAGE) showed that the sizes of the two enzymes were approximately 70 kDa and 30 kDa, in agreement with the calculated theoretical masses (Figure 3), which were also consistent with previous reports [23,24]. Therefore, a biocatalytic synthesis of putrescine from L-arginine as the substrate was successfully engineered.

**Figure 2.** Detection of L-arginine, agmatine, and putrescine. (**a**) HPLC profile of standard putrescine. (**b**) The HPLC profile of the product from the catalytic reaction using the whole-cell catalysis. The retention times of L-arginine, agmatine, and putrescine were 7.4, 17.7, and 29 min, respectively. The sample was collected after the reaction for 30 min.

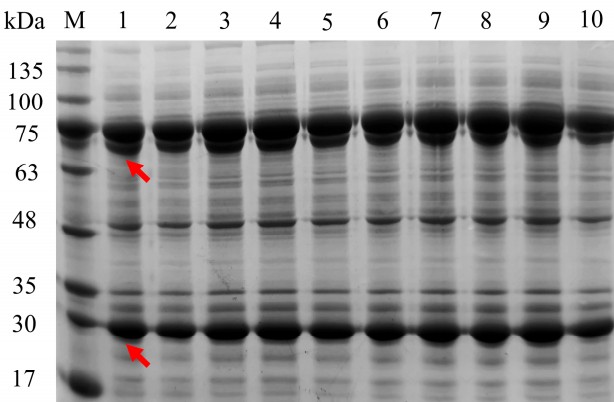

**Figure 3.** SDS-PAGE (12% acrylamide) analysis of ADC (~70 kDa) and AUH (~30 kDa) from strain 1 to strain 10, which were in agreement with the calculated theoretical masses. Lane M: molecular weight markers. The crude enzymes of strain 1 to strain 10 were shown from lane 1 to lane 10, correspondingly. Marker samples of 5 μL and 10 μL were loaded, respectively.

## 2.2. Comparison of Different Strains for Putrescine Productiond

The effects of plasmids with different copy numbers and differently ordered genes on the enzymatic process were compared. The differences in the order of the genes markedly influenced the yield of putrescine. The reason may be that the Duet vector contains dual T7 promoters and dual multiple cloning sites, but only one T7 terminator, which may lead to differences in the transcription levels of the two genes [25].

The effect of plasmids with different numbers of copies on yield was determined (Figure 4). Among the low-copy plasmids (strains 1–6), strain 4 had the highest yield. Although the medium-copy plasmid pETDuet-1 could successfully express both ADC and AUH, the overall catalytic effect of strains 7 and 8 was low. Strain 9 and strain 10 had the highest copy number, but their yields were lower than that of strain 4. The reason may be that the high-copy plasmid combination increases the metabolic pressure and growth burden on the host bacteria [25,26]. The biomass of strains with low-copy plasmids was generally greater than those with high-copy plasmids [27]. Strain 4, containing low-copy plasmids, appeared to have the optimal balance between bacterial growth and protein expression to maximize putrescine production. Therefore, it was selected for all subsequent experiments.

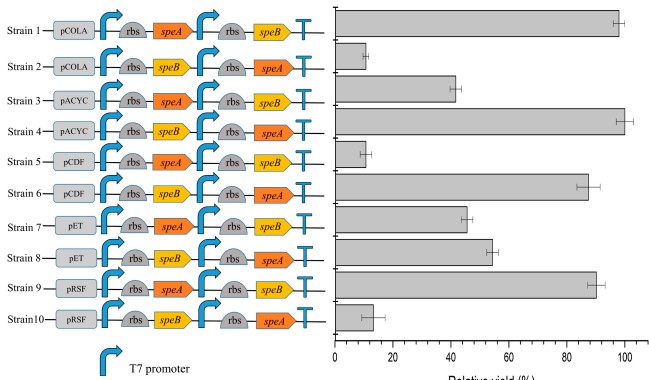

**Figure 4.** Effects of copy numbers of plasmids and gene orders. The *speA* and *speB* encoded ADC and AUH, respectively. The copy number of pCOLADuet-1, pACYCDuet-1, pCDFDuet-1, pETDuet-1, and pRSFDuet-1 was 5, 10, 20, 40, 100, respectively. Standard deviations are indicated by error bars.

## 2.3. Optimization of Whole-Cell Biotransformation Conditions

The effect of pH and temperature on the whole-cell biocatalytic synthesis of putrescine was explored. The optimal biocatalytic temperature of strain 4 was 45 °C (Figure 5a), and its optimal pH was 9.5 (Figure 5b).

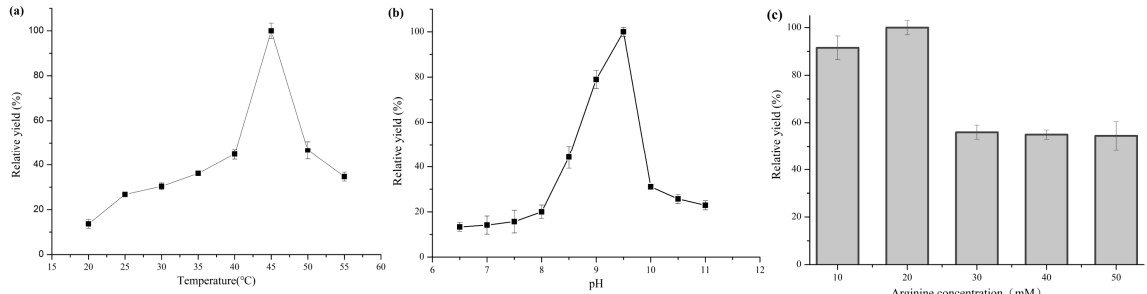

**Figure 5.** Effects of reaction conditions for strain 4. (**a**) Effect of temperature. The pH and L-arginine concentration are 7.5 and 10 mM, respectively. (**b**) Effect of pH (50 mM $KH_2PO_4$-NaOH buffer for pH 6.5–8.0, 50 mM Tris-HCl buffer for pH 8.0–9.0, 50 mM glycine-NaOH buffer for pH 9.5–10.0, KCl-NaOH buffer for pH 11.0). The temperature and L-arginine concentration are 45 °C and 10 mM, respectively. (**c**) Effect of L-arginine concentration. All the experiments were carried out in triplicate. The reaction time is 30 min. Standard deviations are represented by error bars.

The production of putrescine was further studied at different initial concentrations of L-arginine. Under the optimal temperature and pH conditions, the production of putrescine increased until 20 mM L-arginine was reached, then decreased at higher concentrations (Figure 5c). Higher substrate concentration appeared to produce substrate inhibition. Therefore, 20 mM was selected as the optimal initial L-arginine concentration. In summary, the optimal reaction conditions were: 50 mM Tris-HCl, 20 mM L-arginine, pH 9.5, 45 °C, 1 mM pyridoxal-5′-phosphate, 4 mM magnesium sulfate, and 0.1 mM dithiothreitol. The optimized conditions were used in the next experiment.

## 2.4. Time Course of Putrescine Production

Under the optimal reaction conditions, strain 4 was used to determine the time course of putrescine production over 8 h. The conversion increased to 78% rapidly in the first 3 h and then plateaued, presumably because of feedback inhibition by the putrescine product. The conversion only increased by 20% in the subsequent 5 h. The final yield was 98% (Figure 6).

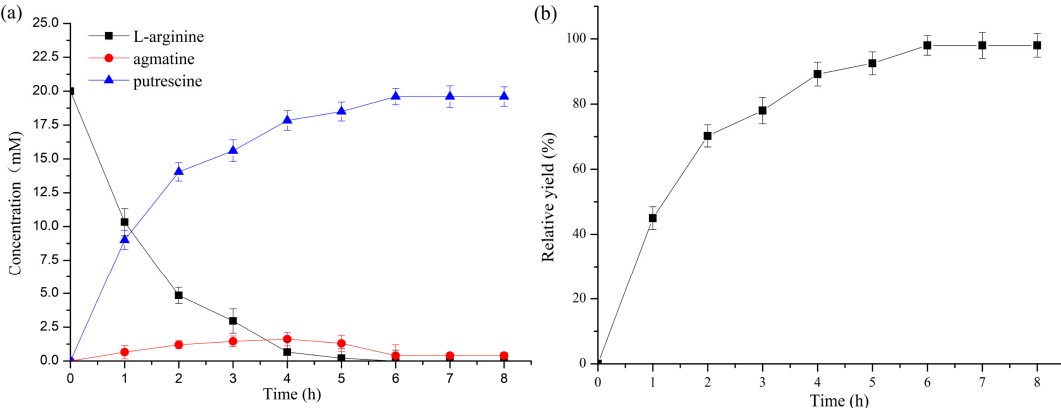

**Figure 6.** Time course of putrescine synthesis from L-arginine by strain 4. (**a**) Concentrations of L-arginine, agmatine, and putrescine with time. (**b**) Mole conversion of putrescine. L-arginine, agmatine, and putrescine were assayed by HPLC. All the experiments were carried out in triplicate. Standard deviations are represented by error bars.

## 3. Discussion

The biosynthetic pathway for putrescine is widely distributed in microorganisms. There are two biosynthetic pathways for the conversion of L-arginine into putrescine in *E. coli*. In Pathway I, L-arginine is converted by arginase into ornithine, then, by ornithine decarboxylase into putrescine. If L-arginine is added exogenously, *E. coli* uses Pathway II, which is the pathway via agmatine, used here (Figure 1) [28]. Both *speA* and *speB* are endogenous genes of *E. coli* and could be expressed effectively.

In the multienzyme cascade reaction system, plasmids with different copy numbers are often used to balance enzyme expression [25,29,30]. In this study, five plasmids with different copy numbers were selected to balance the expression of the two enzymes, and thereby optimize the yield of putrescine. Ten engineered strains were constructed based on different combinations. Strain 4 (pACYCDuet-*speB-speA*) performed the best for the production of putrescine, indicating that low-copy plasmids were better for the production of putrescine than high-copy plasmids. Similarly, in the biosynthesis of N-acetyl-D-neuraminic acid synthesis, the high-copy plasmid pRSFDuet-1 was less efficient than the medium-copy plasmid pCDFDuet-1 [25]. It appeared that, in the selection of plasmid vectors, high-copy plasmids were not always the optimal choice, in some cases, low and medium-copy plasmids performed better in whole-cell biocatalysis.

The optimum temperature of ADC and AUH was 50 °C and 42 °C, respectively [23,24]. As might be expected, the optimal temperature for whole-cell biocatalysis was 45 °C, between the optima of the two enzymes. The optimum pH 9.5 of the biocatalysis was higher than the individual optima of ADC (pH 8.4) and AUH (pH 7.4) [24,31]. This relatively high pH optimum for the overall process may be related to factors other than the enzymes, such as the cellular uptake of L-arginine, or the intracellular transport of putrescine.

Feedback inhibition is common in cellular biosynthetic pathways, as an essential part of their regulation. ADC requires $Mg^{2+}$ as a cofactor and binding of $Mg^{2+}$ to the enzyme is competitively inhibited by both L-arginine and putrescine [31]. Therefore, 4 mM $Mg^{2+}$ was added to the reaction buffer to minimize feedback inhibition of cofactor binding. The biocatalytic process had an optimal L-arginine concentration of 20 mM, indicating that feedback inhibition occurs at higher concentrations, in agreement with the previous report [31]. Although the increase in $Mg^{2+}$ concentration reduced the inhibition of putrescine towards ADC, putrescine was still inhibited at high concentrations [31]. There is a complex relationship between the inhibitors and cofactors of ADC, which could be modified by protein engineering, to relieve the feedback inhibition by the biogenic amine [32]. AUH is not inhibited by putrescine, but L-arginine competitively inhibits AUH ($Ki = 9 \times 10^{-3}$ M) [24]. AUH would also benefit from protein engineering, to relieve the inhibitory effect of L-arginine to enhance enzyme activity. The accumulation of the intermediate product agmatine was observed during the cascade reaction (Figure 6a), which does not affect the overall catalytic process. Protein engineering may produce an AUH with higher catalytic activity.

The ornithine decarboxylase (ODC) pathway, with glucose as the substrate, was enhanced in *E. coli*, achieving a yield 24.2 g $L^{-1}$ after a high-density culture for 32 h [10]. In 2012, the yield reached 19 g $L^{-1}$ after high-density fermentation using *C. glutamicum* for 34 h [14]. This work started from L-arginine to produce putrescine by over-expressing two enzymes. After centrifugation, the strains were resuspended with buffer and kept in resting state in the process of catalysis. Other enzymes that may produce side reactions have lower content compared to these two enzymes. The whole-cell biocatalytic process was efficient because of the high permeability of *E. coli* to the substrate and product, and the high intracellular enzyme concentration. The 0.2 g $L^{-1}$ of dry cell weight in the flask could make the reaction quick and effective, with a short metabolic pathway.

In this study, a whole-cell biocatalytic system was constructed that used L-arginine as the substrate to produce putrescine through dual-enzyme catalysis. L-arginine is a cheap and readily available raw material, which can be converted relatively easily into high-value putrescine [33]. This biosynthetic method has the advantages of simple operation, low cost, and high yield. In the future, industrial production of putrescine could be improved through the in-situ separation of products or molecular

evolution. Compared with traditional separation methods, in-situ separation methods can reduce the feedback inhibition of products and the use of organic reagents, which is of great significance for the industrial production of putrescine. Based on the characteristics of putrescine, materials like resin could be designed to separate putrescine from the culture broth to improve the yield and fermentation efficiency. The methods of in-situ separation include adsorption resin, ion exchange resin, foam chromatography, ultrafiltration, and so on. At present, it is meaningful and challenging to establish the method of in-situ separation of putrescine, and match well with *E. coli* whole-cell catalysis [34].

## 4. Materials and Methods

### 4.1. Strains, Plasmids, Other Materials

*E. coli* JM109 and *E. coli* BL21 (DE3) were used as the cloning host and the expression host, respectively. The plasmids pACYCDuet-1, pCOLADuet-1, pCDFDuet-1, pETDuet-1, and pRSFDuet-1 were from Novagen (Darmstadt, Germany). The DNA gel extraction kit, plasmid miniprep kit, and Taq DNA polymerases were from TaKaRa. The multiF seamless connection kits were from Abclonal (Wuhan, China). The isopropyl β-D-1-thiogalactopyranoside, ampicillin, kanamycin, chloramphenicol, and streptomycin were from Sangon Biotech (Shanghai, China). Standards: the L-arginine, agmatine sulfate, putrescine, and pyridoxal-5'-phosphate were from Aladdin (Shanghai, China). The methanol and tetrahydrofuran were chromatography grade, from Tedia (Fairfield, OH 45014, U.S.A.) and Rhawn (Shanghai, China), respectively. Primer synthesis and gene sequencing were performed by Talen-bio Scientific (Shanghai, China). The primers used are listed in Table 1.

**Table 1.** Primers used in the study.

| Primers | Sequences, 5'-3' |
|---|---|
| Site1-*speA*-F | TTAAGTATAAGAAGGAGATATACATATGAGCACCTTAGGTCATCAATACG |
| Site1-*speA*-R | TTAAGCATTATGCGGCCGCAAGCTTTTACTCATCTTCAAGATAAGTATAAC |
| Site1-*speB*-F | TCACCACAGCCAGGATCCGAATTCGATGAGCACCTTAGGTCATCAATACG |
| Site1-*speB*-R | TTTCTTTACCAGACTCGAGGGTACCTTACTCGCCCTTTTTCGCCG |
| Site2-*speA*-F | TTAAGTATAAGAAGGAGATATACATATGTCTGACGACATGTCTATGGGT |
| Site2-*speA*-R | TTAAGCATTATGCGGCCGCAAGCTTTTACTCGCCCTTTTTCGCCGC |
| Site2-*speB*-F | TTAAGCATTATGCGGCCGCAAGCTTTTACTCATCTTCAAGATAAGTATAAC |
| Site2-*speB*-R | TTTCTTTACCAGACTCGAGGGTACCTTACTCATCTTCAAGATAAGTATAACC |

### 4.2. Pathway and Plasmid Construction

Using *E. coli* BL21 (DE3) as the template, *speA* (GenBank ID: CP032667.1) and *speB* (GenBank ID: CP028306.1) were amplified by PCR, encoding ADC and AUH, respectively. Plasmids with different copy numbers were selected for comparison: pCOLADuet-1 (low-copy, copy number 5), pACYCDuet-1 (low-copy, copy number 10), pCDFDuet-1 (low-copy, copy number 20), pETDuet-1 (medium-copy, copy number 40), and pRSFDuet-1 (high-copy, copy number 100) [21]. The genes were ligated to the plasmids with multiF seamless connection kits. For each plasmid, two versions with *speA* and *speB* in different orders were constructed. The plasmid was transformed into *E. coli* BL21 (DE3), and the success of the transformation was confirmed by both the presence of the antibiotic resistance gene and by gene sequencing. The plasmids and strains used in this study are listed in Table 2.

**Table 2.** Strains and plasmids used in this study.

| Strains/Plasmids | Description | Source |
|---|---|---|
| Plasmids | | |
| pCOLADuet-1 | double T7 promoters, COLA ori, KanR | Novagen |
| pACYCDuet-1 | double T7 promoters, P15A ori, ChlR | Novagen |
| pCDFDuet-1 | double T7 promoters, CDF13 ori, SmR | Novagen |
| pETDuet-1 | double T7 promoters, pBR322 ori, AmpR | Novagen |
| pRSFDuet-1 | double T7 promoters, RSF ori, KanR | Novagen |
| pCOLADuet-*speA-speB* | pCOLADuet-1 carrying *speA* and *speB* | this study |
| pCOLADuet-*speB-speA* | pCOLADuet-1 carrying *speB* and *speA* | this study |
| pACYCDuet-*speA-speB* | pACYCDuet-1 carrying *speA* and *speB* | this study |
| pACYCDuet-*speB-speA* | pACYCDuet-1 carrying *speB* and *speA* | this study |
| pCDFDuet-*speA-speB* | pCDFDuet-1 carrying *speA* and *speB* | this study |
| pCDFDuet-*speB-speA* | pCDFDuet-1 carrying *speB* and *speA* | this study |
| pETDuet-*speA-speB* | pETDuet-1 carrying *speA* and *speB* | this study |
| pETDuet-*speB-speA* | pETDuet-1 carrying *speB* and *speA* | this study |
| pRSFDuet-*speA-speB* | pRSFDuet-1 carrying *speA* and *speB* | this study |
| pRSFDuet-*speB-speA* | pRSFDuet-1 carrying *speB* and *speA* | this study |
| Strains | | |
| strain 1 | *E. coli* BL21 (DE3)/pACYCDuet-*speA-speB* | this study |
| strain 2 | *E. coli* BL21 (DE3)/pACYCDuet-*speB-speA* | this study |
| strain 3 | *E. coli* BL21 (DE3)/pCOlADuet- *speA-speB* | this study |
| strain 4 | *E. coli* BL21 (DE3)/pCOlADuet- *speB-speA* | this study |
| strain 5 | *E. coli* BL21 (DE3)/pETDuet- *speA-speB* | this study |
| strain 6 | *E. coli* BL21 (DE3)/pETDuet- *speB-speA* | this study |
| strain 7 | *E. coli* BL21 (DE3)/pCDFDuet- *speA-speB* | this study |
| strain 8 | *E. coli* BL21 (DE3)/pCDFDuet- *speB-speA* | this study |
| strain 9 | *E. coli* BL21 (DE3)/pRSFDuet- *speA-speB* | this study |
| strain 10 | *E. coli* BL21 (DE3)/pRSFDuet- *speB-speA* | this study |

### 4.3. Culture Conditions and Preparation of Whole-Cell Biocatalysts

The seed solution was activated overnight in test tubes containing Luria−Bertani (LB) medium (3 mL), then added to 50 mL LB medium at a ratio of 1:50 (v/v). Various antibiotics were added at the start of the culture, including ampicillin (100 mg mL$^{-1}$), kanamycin (40 mg mL$^{-1}$), chloramphenicol (20 mg mL$^{-1}$), and streptomycin (40 mg mL$^{-1}$). The strains were incubated for 2 h in 250 mL shake flasks at 37 °C and 200 rpm. When the optical density reached 0.6 at 600 nm, isopropyl β-D-1-thiogalactopyranoside was added to a final concentration of 0.4 mM. Subsequently, strains were incubated, in shake flasks for 24 h at 200 rpm and 15 °C, and the cells were then collected by centrifugation (8000× *g*, 4 °C, 10 min). Finally, they were resuspended in the buffer. *E. coli* BL21 (DE3), but without the target genes, was used as the control.

### 4.4. SDS-PAGE Analysis

The cells were disrupted with an ultrasonic probe for 20 min and centrifuged at 5000× *g* for 10 min. Then, the denaturing buffer (10 μL) was added to the supernatant (40 μL) and heated for 10 min at 100 °C. SDS-PAGE (12% acrylamide) was used for electrophoresis. After electrophoresis, the protein bands were stained with Coomassie Brilliant Blue R250. The bands were imaged using BandScan software version 4.3, after destaining.

### 4.5. Whole-Cell Biocatalysts and Optimization of Reaction Conditions

The reaction was carried out in a buffer and included the following components: 20 mM L-arginine, 4 mM magnesium sulfate, 1 mM pyridoxal-5′-phosphate, and 0.1 mM dithiothreitol. The reaction was carried out in a 250 mL shake flask, at 37 °C and 200 rpm. The buffer was used to adjust the cell density of each strain to the same level. Then, the same amount of each strain was used for the

whole-cell catalysis trials. To start the reaction, bacterial cells (10 mg dry weight) were added to the reaction solution (50 mL) for 30 min. Finally, 1/5 volume of trichloroacetic acid was added to terminate the reaction.

To evaluate the effect of temperature on biotransformation, the reaction was carried out at 20, 30, 40, 50, and 60 °C. In order to evaluate the influence of pH, the pH of the reaction system was adjusted to 6.5, 7.0, 7.5, 8.0, 8.5, 9.0, 9.5, 10.0, 10.5, or 11.0 (50 mM $KH_2PO_4$-NaOH buffer for pH 6.5–8.0, 50 mM Tris-HCl buffer for pH 8.0–9.0, 50 mM glycine-NaOH buffer for pH 9.5–10.0, KCl-NaOH buffer for pH 11.0). Different concentrations of L-arginine (5, 10, 20, 30, and 40 mM) were used to determine the optimal substrate concentration. All reactions were carried out in triplicate.

### 4.6. Analytical Methods

The concentrations of L-arginine, agmatine, and putrescine were determined by HPLC, as reported previously, with minor modifications [35]. The instrument was an Agilent 1260 Infinity (Agilent Technologies, Santa Clara, CA, USA), fitted with a Waters (Milford, MA, USA) Sunfire C18 column ($4.6 \times 250$ mm, 5 μm), and an Agilent 1260 fluorescence detector G1321C. The column temperature was 25 °C, the flow rate was 1 mL min$^{-1}$, and the injection volume was 10 μL. The mobile phases were: Solvent A-0.050 M acetate buffer/tetrahydrofuran (96/4, pH 6.0) and Solvent B-methanol. The percentage of Solvent B was 0% from 0 to 17 min, ramped linearly to 33.3% at 22 min, then to 100% at 30 min, and maintained at 100% until 35 min. The excitation and emission wavelengths were 340 nm and 420 nm, respectively. The O-phthaldialdehyde reagents were from Agilent Technologies.

**Author Contributions:** Conceptualization, H.H. and Y.B.; methodology, T.-P.F.; software, H.H.; validation, Y.C. and T.-P.F.; formal analysis, H.H.; investigation, H.H.; resources, X.Z.; data curation, Y.C.; writing—original draft preparation, H.H.; writing—review and editing, H.H. and Y.C.; visualization, T.-P.F.; supervision, X.Z.; project administration, X.Z.; funding acquisition, Y.C. All authors have read and agreed to the published version of the manuscript.

**Funding:** We thank the National Key Scientific Instrument and Equipment Development Project of China (2013YQ17052504), the Program for Changjiang Scholars and Innovative Research Teams in the University of Ministry of Education of China (IRT_15R55), the Natural Science Foundation of Shanxi province (2019JQ-725), and the Postgraduate Research & Practice Innovation Program of Jiangsu Province (KYCX19_1841, KYCX19_1842) for financial support.

**Conflicts of Interest:** The authors declare no conflict of interest.

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
