# Peer review of "Biosynthesis of Putrescine from L-arginine Using Engineered Escherichia coli Whole Cells"

_catalysts, doi:10.3390/catal10090947_

Round 1

Reviewer 1 Report

The work presents an innovative putrescine synthesis using engineered E. coli cells and L-arginine as a substrate. A highly systematic study is well performed and shows promising results.

There are just a few minor remarks which might improve the manuscript:

1. In Chapter 2.3 and in Fig.5, the time of the process could be defined. Besides, the pH value and substrate concentration used in Fig.5a, as well as the temperature and substrate concentration used in Fig.5b might be specified.

  1. The terminology “feedback inhibition” in Chapters 2.4 and 3 is somehow vague, as the authors and the literature state that L-arginine competitively inhibits AUH (p. 5, line 160) and that there is an inhibitory effect of L-arginine (p. 5, line 161). Therefore, substrate inhibition might be considered. However, Figure 5c implies that higher (>30 mM) arginine concentrations do not lead to lower product yields, but rather gives similar yield. Furthermore, Fig. 6 shows typical lowering of biocatalytic reaction rate by time due to lower substrate concentration, so considering 98 % yield it is hard to claim that there is evidence of substrate inhibition. Further discussion on feedback inhibition on p.5 also claims the inhibition above 20 mM substrate concentration and not at the conditions used in Fig. 6. It would be great to show the reaction initial rate vs. substrate concentration (could be calculated from the data used in Fig. 5c?) in order to prove the substrate inhibition.
  2. Fig. 6b is the only one using conversion instead of yield. It would be worth unifying the results.
  3. The authors ambitiously claim the possibility of industrial production of putrescine by the improved biocatalysts used in the study and in situ separation. Could they envisage the continuous process with the long-term use of E. coli cells?
  4. A small typo improvement: de novo and in situ should be written in italic.

Reviewer 2 Report

This paper describes the development of a new E.coli strain for the efficient production of putrescine from L-arginine by whole cell biocatalysis expressing two different enzymes. The introduction is well structure and includes adequate references to the recent developments for ornithine production.

I just have a few experiments suggestions to enhance the quality of the manuscript that the authors might consider:

  • About the use of multienzymatic systems in the introduction, recent references might be added, as the referenced is only a review (Ref20).
  • The authors have tried to use different plasmid with different copy number to find the best system, but from the time course production, it seems that the decarboxylation is the limiting step, as no more than 2.5 mM of agmatine is detected at any point. Have the authors tried cotransformation where the two genes are in different plasmids? Maybe this would help optimise the ratio between both enzymes and therefore, increase the productivity of the sytem.
  • Could the authors explain why concentrations of more than 20 mM arginine did not translate with higher productions? Did they see any growth inhibition or side effects of the higher concentration? They mention in the next part feed-back inhibition, and comment on that also on the discussion, but maybe a sentence could be added there.
  • Although the production is 100% at 20 mM, at 50 mM a higher yield is obtained. Also, have the authors tried a fed batch method, reusing the same cells? If L-arginine is added step-wise, the productivity of the system could be greatly enhanced.
  • “The accumulation of the intermediate product agmatine was observed, during the cascade reaction. (Figure 6a), which may result from insufficient expression of AUH or low AUH enzyme activity.”
    No accumulation of the intermediate is really seen in my opinion. It seems that the AUH enzyme works very efficiently, converting all agmatine formed to putrescine as the decarboxylase produces it.
  • Line 73: Reword to make clear the low concentration of cells. The sentence is somewhat unclear.

Reviewer 3 Report

The authors describe the biosynthesis of putrescine from L -arginine using 3 engineered Escherichia coli whole cells. The manuscript is very well written with a classical format for a publication in Catalysts an appropriate introduction, a part devoted to the results, another to the discussion and an experimental part. The small conclusion is included at the end of the discussion part and revealed also some perspectives for further developments. The figures, schemes and tables are clear and well realized.

The introduction is devoted first to the putrescine, a biogenic amine and its various applications (biology, agriculture, material,..). Then different biological synthesis methods have been listed to explain the benefit of the research presented in this manuscript. The literature concerning these two parts is quite complete with old and recent references.

A whole-cell biocatalytic method in Escherichia coli was explored for the production of putrescine, using L-arginine as the substrate. Indeed, L-arginine decarboxylase and agmatine ureohydrolase were co-expressed to produce putrescine from L-arginine. The authors used 10 plasmids with different copy numbers and ordering of genes; the expression of the two enzymes has been explored and the best strain was pACYCDuet-speB-speA.

The screening was carefully realized and the optimal concentration of L-arginine was also determined as 20 mM using pACYCDuet-speB-speA. The study was next performed by the determination of the best time, pH and temperature conditions. Here again, the work is rigorous and led to suitable conditions for the production of putrescine with excellent yields.

In conclusion, in this form, I’m able to recommend this manuscript for a publication in Catalysts.
